# Influence of Si and SiC Coating on the Microstructures and Mechanical Properties of C/C Bolts

**DOI:** 10.3390/ma16051785

**Published:** 2023-02-21

**Authors:** Guodong Sun, Zhiqiang Tan, Qing Zhang, Yi Zhang, Xuqin Li, Qinglai Tian, Yuxing Tang

**Affiliations:** 1School of Materials Science and Engineering, Chang’an University, Xi’an 710064, China; 2Science and Technology on Thermostructural Composite Materials Laboratory, Northwestern Polytechnical University, Xi’an 710072, China; 3School of Materials and Environmental Engineering, Chengdu Technological University, Chengdu 611730, China

**Keywords:** vapor silicon infiltration, carbon/carbon composite, bolt, mechanical property

## Abstract

High−performance bolts made of carbon/carbon (C/C) composites are necessary for connecting thermally−insulating structural components of aerospace vehicles. To enhance the mechanical properties of the C/C bolt, a new silicon infiltration−modified C/C (C/C−SiC) bolt was developed via vapor silicon infiltration. The effects of silicon infiltration on microstructure and mechanical properties were systematically studied. Findings reveal that dense and uniform SiC−Si coating has been formed after silicon infiltration of the C/C bolt, strongly bonding with the C matrix. Under tensile stress, the C/C−SiC bolt undergoes a tensile failure of studs, while the C/C bolt is subject to the pull−out failure of threads. The breaking strength of the former (55.16 MPa) is 26.83% higher than the failure strength of the latter (43.49 MPa). Under double−sided shear stress, both the crushing of threads and the shear failure of studs occur within two bolts. As a result, the shear strength of the former (54.73 MPa) exceeds that of the latter (43.88 MPa) by 24.73%. According to CT and SEM analysis, matrix fracture, fiber debonding, and fiber bridging are the main failure modes. Therefore, a mixed coating formed by silicon infiltration can effectively transfer loads from coating to carbon matrix and carbon fiber, thereby enhancing the load−bearing capacity of C/C bolts.

## 1. Introduction

Carbon/carbon (C/C) composites possess outstanding properties, such as high specific strength and stiffness, good resistance to heat, chemical corrosion, thermal shock, low coefficient of thermal expansion, and high strength retention at elevated temperatures. Exhibiting strength reduction in the inert environment over 2000 °C, carbon materials have been applied in spacecraft’s thermal protection systems, nozzles of solid rocket motors, brakes of aircraft and racing cars, heat exchangers, heat−dissipation systems of high−power electronic devices, etc. [1]. Constraints on the size of manufactured components usually require mechanical connecting methods for the assembly and shaping of large−size structural components [2,3,4]. In particular, the bolt connection allows the disassembly of the structure to inspect and re−use of structure, so it is the common connection method for large−size structural components [5,6,7]. Taking the re−entry of a spacecraft into the atmosphere as an example, the maximum temperature of the C/C flap can exceed 1650 °C [8] due to pneumatic heating. Thus, developing new C/C bolts with high temperature resistance and remarkable mechanical properties is an urgent task.

The performance of C/C bolts depends on the mechanical properties of bulk C/C composites, which are closely related to the types of carbon fibers, fiber architecture, and preparation processes. PANG Sheng−Yang investigated the effect of preform process parameters on the mechanical properties of C/C bolts with three−dimensional needle−punched carbon fibers. The investigation of 3D needle−punched C/C composites [8] showed that the tensile and bending strength of C/C increased as the density of the needle−punched decreased. As the carbon yarn bundle varied from 12 K to 3 K, the tensile bending strength of C/C also increased. Therefore, the structurally optimized 3D needle−punched preforms have become the preferred fiber architecture for developing C/C bolts. To enhance the oxidation resistance of C/C bolts, a SiC coating was prepared on the surface of C/C bolts by chemical vapor deposition in the study [9], which enabled one to form SiC−modified C/C (C/C−SiC) bolts. The modification enhanced the interfacial bonding strength between the fibers and the matrix under the air environment at 1100 °C. H. Böhrk [5] proposed a new type of C/C−SiC safe fastening bolt whose feasibility and technical advantage was verified via the re−entry flight examination. Thus, the above researches indicate that SiC coating modification can strengthen and toughen C/C bolts. One reason is that the tensile strength of C/C bolts is slightly lower than that of C/C composites due to processing defects. In this respect, the introduction of SiC coating can repair the processing damage of C/C bolts, reinforcing them. The other reason is that the fiber fracture and matrix stripping are the main failure mechanisms of the C/C bolted joints [10]. Applying SiC coating can reduce porosity and further improve the load transfer between the fiber and matrix.

However, the preparation process, micro−structural characteristics, mechanical properties, and failure mechanisms of the two bolts still need further study to elucidate the strengthening and toughening mechanisms during the SiC coating modification.

In view of the above, SiC and Si coatings were formed on the surface of C/C bolts via vapor silicon infiltration. The C/C bolts themselves were preliminarily produced using a 3D needle−punched prefabricated structure. Special attention was paid to the effects of silicon infiltration modification on the mechanical properties of C/C bolts. The microstructure of C/C bolts was analyzed via CT, SEM, and profilometry. The constituent phases before and after modification were determined by EDS, Raman spectroscopy, and XRD. Finally, the tension and double−sided shear failure mechanisms of the C/C−SiC bolts were discussed.

## 2. Materials and Methods

### 2.1. Preparation of C/C and C/C−SiC Bolts

As shown in Figure 1a, CH_4_ (99%) and ethanol (99%) with flow rates of 25 μL/min and 2048 μL/min, respectively, were introduced as the mixed carbon sources at 1150 °C and 5 KPa. The 3D needle−punched carbon fiber was densified to 1.3–1.4 g/cm^3^ by chemical vapor infiltration [11] (CVI) for 200 h. Precursor infiltration pyrolysis (PIP) was afterward implemented to further increase the density of the fiber to 1.5 g/cm^3^. For this, the fiber was impregnated with furfural ketone resin, cured at 120–180 °C, and finally heated at 950 °C under vacuum [12]. A zero orientation angle of the fiber bundle was chosen as the axial direction of the bolt. The C/C composite material was machined into M10 hexagon−head bolts, which were then exposed to the CVI process for 10 h to seal the holes therein.

Figure 1b depicts the 3D needle−punched carbon fiber that was first densified to 0.8–1.0 g/cm^3^ through CVI for 100 h and then to 1.5 g/cm^3^ by a triple cyclic PIP process. The distribution direction of a 0° fiber bundle was the bolt’s axial direction, and the C/C composite material was machined into M10 hexagon−head bolts. After that, M10 C/C bolts were placed in an airtight graphite crucible, and silicon powder in the quantity of 1.5 times the mass of the bolt was added. The mixture was kept at the temperature of 1700 °C for 2 h to prepare the C/C−SiC bolts by vapor silicon infiltration [13].

### 2.2. Measuring the Mechanical Properties

The tension (Figure 2a) and double−sided shear properties (Figure 2b) of the bolt were measured using a hydraulic servo electronic universal tester (Changchun DNS−100) at a speed of 0.5 mm/min according to tensile and double−sided shear test standard (ASTM D5961). The calculation equations for strength are as follows:(1)σ=F/A
(2)τ=F/2A
where *A* denotes the cross−sectional area of the M10 standard bolt (58.00 mm^2^); *F* stands for the external force load (N); *σ* is the tensile strength (MPa), *τ* denotes the double−sided shear strength (MPa). In this paper, we study C/C and C/C−SiC bolts and therefore use metal nuts in conjunction with them. It is guaranteed that mechanical failure occurs on C/C and C/C−SiC bolts.

### 2.3. Observation of Microscopic Structures

The surface morphology of the bolts was observed using a profiler (Nanovae ST400, CA, USA) and a scanning electron microscope (SEM, SU3800/SU4800, Hitachi, Japan). The microstructural analysis of the sample was performed via CT (AX−2000, Ningbo Auger Detection).

### 2.4. Phase Composition Analysis

An X−ray diffractometer (XRD, Bruker−D8 FOCUS) equipped with a copper target was used to analyze the phase composition via continuous scanning at 40 kV and 10 mA. The XRD spectrograms were recorded in a scanning angle range of 5–80° at the exposure time of 10 s and a scanning speed of 5°/min. The Raman spectroscopy (LabRAM, Renishaw in Via Reflex) at a 514 nm laser excitation wavelength was applied to analyze the phase composition. Prior to the measurements, samples were taken from the surfaces of C/C and C/C−SiC threads and ground into powders with an agate crucible. The elemental analysis of the C/C−SiC bolt phase was conducted using an energy dispersive spectrometer (EDS, Xplore15).

## 3. Results and Discussion

### 3.1. Microstructure and Phase Composition

Figure 3a shows XRD images of C/C and C/C−SiC threaded tooth powders. Figure 3b shows the Raman images of C/C and C/C−SiC threaded tooth powders. By comparison with literature 14–16, it can be seen that the C/C−SiC bolts contain three phases of C, Si, and SiC after the silica modification. [14,15,16].

Figure 4a shows the EDS of the C/C−SiC coated bolts, where the three phases of C, Si, and SiC can be clearly distinguished. This indicates that the Si and SiC produced by the vapor phase silicon infiltration are mixed together on the surface of the C/C bolt. Figure 4b shows the surface morphology of the C/C−SiC bolt, which clearly shows that the SiC and SiC are tightly packed together in a granular form wrapped around the C/C bolt. Figure 4c depicts the CT image of the modified C/C−SiC threads. The C/C bolts covered with the coating were removed from the picture. The picture image exhibited a 30 μm thin and uniformly distributed coating, which was tightly bonded to the C/C bolt through the pinning effect. There was no obvious interface between the coating and the matrix, which indicated an improvement in the compatibility and bonding performance of the two components.

Figure 5a shows the contour of a C/C bolt, and Figure 5b shows the contour of a C/C−SiC bolt. It is seen that the bolts are standard M10 hexagonal convex head bolts. The surface roughness values of both bolts were quite close (67.46 μm and 62.09 μm), respectively), indicating that vapor silicon infiltration did not affect the roughness of C/C bolts.

As can be seen in Figure 6a,c, C/C bolts possessed the 3D needle−punched fiber architecture, in which 0° fiber bundles were distributed along the axial direction of bolts, while 90° fiber bundles and tire webs were perpendicular to the axial direction of bolts. There are a large number of interlayer cracks elongated among 0° fiber bundles, 90° fiber bundles, and tire webs, as well as the needle−punched (round−like) pores between the needle−punched positions and weft−free fiber bundles. There are also abundant internal pores, causing C/C threads to be easily crushed and to fall out during machining.

After the C/C bolts were modified by silicon infiltration (Figure 6d), gas silicon was put into the inner part of the 3D needle−punched prefabricated body. It can be seen with the naked eye that the above cracks have been all filled, while some closed needle−punched pores remain empty. Figure 7 shows the pore volume distribution of bolts along the *X*−axis. According to the plot, the minimum pore volume (9 × 10^3^ mm^3^) of C/C bolts was larger than the maximum pore volume (8 × 10^3^ mm^3^) of C/C−SiC bolts, indicating that vapor silicon infiltration could effectively reduce the porosity of C/C bolts. As seen in Figure 6b, due to the thermal stress caused by high−temperature silicon infiltration at 1700 °C, numerous short periodic cracks occurred in the interlayer between the 90° fiber bundles, and scarce long interlayer cracks appeared between the 0° and 90° fiber bundles. The phenomenon resulted from the difference in thermal expansion coefficients between the silicon infiltration matrix and C/C composite.

On the other hand, a SiC−Si coating was formed on the threaded surface (Figure 6d), which effectively hindered the crack extension caused by stress concentration and prevented the damage of threads in machining. The complete threads were conducive to enhancing the comprehensive mechanical properties of bolts. Figure 4c depicts the CT image of the modified C/C−SiC threads. C/C−SiC threads covered with the coating were removed from the picture. The picture image exhibited a 30 μm thin and uniformly distributed coating, which was tightly bonded to the C/C bolt through the pinning effect. There was no obvious interface between the coating and the matrix, which indicated an improvement in the compatibility and bonding performance of the two components.

### 3.2. Mechanical Properties

#### 3.2.1. Tensile Properties

Figure 8a,b show the tensile load−displacement curves of C/C and C/C−SiC bolts, respectively. The load of the C/C bolt rose slowly and non−linearly before being pulled 0.2 mm. Afterward, the load rises increased linearly and dropped sharply after reaching its maximum load. The linear curve indicates that a brittle fracture has occurred, and the tensile fracture surface shows that the thread tooth is pulled out. The tensile load−displacement curve of the modified C/C−SiC bolt has not changed significantly compared with that of the unmodified C/C−SiC bolt, but the tensile fracture surface reveals that the C/C−SiC bolt has undergone a stud tensile fracture. Table 1 summarizes the tensile mechanical properties of C/C and C/C−SiC bolts. The tensile strengths of C/C and C/C−SiC bolts are 43.49 MPa and 55.16 MPa, respectively, showing a 26.83% increase after silicon infiltration. The fracture displacement of the C/C bolt is 0.5 mm, and that of the silicon−modified bolt is 1.0 mm, meaning a 100% increase after silicon infiltration. Thus, the silicon infiltration modification can enhance the fracture strength and displacement of C/C bolts simultaneously.

#### 3.2.2. Double−Sided Shear Properties

Figure 8c,d depict the double−sided shear−load displacement curves of C/C and C/C−SiC bolts. Both curves exhibit a bimodal shear profile, which is attributed to the fact that thread crushing and stud shearing breakage successively occur as the bolts are subjected to double−sided shear. When the displacement of the C/C bolt is lower than 1.2 mm, the load first rises linearly until its first maximum and then slowly decreases. After that, it increases to the second maximum until the bolt breaks. The analysis of the double−sided shear fracture surface revealed that the C/C bolt suffered from thread crushing and stud shearing failure. The double−sided shear load−displacement curve did not change significantly after silicon infiltration modification. Moreover, the fracture observation showed that the C/C−SiC bolt also experienced crushing of threads and shear breakage of studs. Table 1 lists the shear properties of C/C and C/C−SiC bolts. The crushing load of C/C threads is 3518.67 N, and that of C/C−SiC threads is 3770.00 N, showing a 7.16% increase after silicon infiltration. The double−sided shear strength of the C/C bolt is 43.88 MPa, and that of the C/C−SiC bolt is 54.73 MPa, i.e., there was an increase of 24.73% after silicon infiltration. The fracture displacement of the C/C bolt is 2.5 mm, and that of the modified bolt is 1.7 mm. In a word, the fracture displacement of the C/C bolt was reduced by 32% after silicon infiltration.

Therefore, both the bearable crushing load of threads and the double−sided shear fracture load of studs increased after silicon infiltration modification. In addition, the crushing rigidity and shear−fracture rigidity were also enhanced.

### 3.3. Failure Mechanisms

#### 3.3.1. Tensile Failure

Figure 9 shows the SEM micrographs, revealing the tensile failure of C/C and C/C−SiC bolts. During the densification process of C/C composites, the C matrix produced by the CVI and PIP processes was mainly distributed between 0° and 90° fiber bundles, but the tire’s reticulum layer was relatively loose. Meanwhile, the introduction of puncture fiber bundles resulted in needle−punching voids which were not filled by the C matrix. This damaged the 0° fiber bundles in threading. When C/C bolts were subjected to tensile loading, 0° fiber bundles of the thread and the C matrix first slipped, and then cracks extended along the fiber/matrix interface, finally resulting in the interfacial debonding between the 0° fiber bundle and the C matrix (Figure 9b). However, the 90° fiber bundle was subjected to shear stress and finally fractured at the defects, manifested by the pull−out failure of threads (Figure 9a).

The tensile stress from the thread was transferred to the studs through the coating. Inside the C/C−SiC studs, cracks were generated around the pores between the 90° fiber layer perpendicular to the tensile direction and the tire web. The cracks then propagated along the interlayer pore in the 90° direction, leading to the fracture of the C matrix. The 90° fiber bundle and the interfacial layer of the matrix were both subjected to tensile stress perpendicular to the interfacial layer, which caused the tear and failure of the latter one. There 90° fibers are debonded (Figure 9e). Besides, porosity existed between the 0° and 90° fiber bundles. Under tensile loading, stress was concentrated in the pores around the C matrix, and cracks spread among the fiber bundles. The cracks further extended along the inter−laminar pores in the 0° fiber bundles until bundles were pulled out and broken, presenting the fracture of studs (Figure 9d,e). The transition from the pull−out failure mode of threads to the fracture failure of studs reflected the reinforcing effect provided by the coating (Figure 9a,d).

A Z−direction fiber “pin” action formed by needle−punched fiber bundles (Figure 9c,f) intensified the stress transfer between different layers. However, the fiber layer failed to bear the load since the directions of carbon fibers at the “pin” positions were perpendicular to the axial tensile stress, mainly manifested due to the stress within the C−matrix. Therefore, once the cracks propagated at the interfacial interface, the “pin” tended to vary the direction of cracks, making them tear the pin and extend down to the next interfacial layer. Under tensile stress, 0° fiber bundles underwent the fracture and pull−out, while cracks formed between 90° fiber bundles and the matrix, leading to interfacial debonding. Therefore, the tensile properties of C/C composite bolts are determined by the thread’s bearing capacity and the tensile properties of the C/C material itself (Figure 10a,b)

#### 3.3.2. Double−Sided Shear Failure

As seen in Figure 11a,d, C/C threads successively experienced crushing of threads and double−sided shear failure of studs under engineering shear stress. In particular, matrix cracking occurs on the surface of C/C and C/C−SiC bolts under compressive stress, producing cracks (Figure 11b,e). In turn, studs were shear−fractured in the shear plane of the bolts. Therefore, the main damage mechanism was the fracture failure of the 90° fiber bundles induced by double−sided shear stress. For C/C and C/C−SiC bolts under double shear loading, 0° fiber bundle shear, fiber pull−out, fracture, 90° fiber bundle tension, and interfacial debonding occur (Figure 11c,f).

According to the above fracture morphology analysis, the failure mechanism of double−sided shear−induced fracture of studs was summarized as follows. As seen in Figure 12, studs were subjected to axial pressure, and shear stress occurred at the middle line of the interfacial layer. Because of the mechanical transmission of Z−direction fibers, the unit layers at both sides of the center line were subjected to compressive stress. Since the weft−free fiber bundles in the unit layer were arranged at different angles, there were certain differences in the fracture morphology of the interfacial layer. Specifically, the bending of fiber bundles at 90° resulted in matrix cracking between the fibers. Moreover, fiber debonding occurred between the 90° fiber bundle and the C matrix. In addition, the cracks extended along the 0° fiber bundle, and fiber bridging with subsequent fracture and pull−out of the bundle was observed. Thus, the coating enhanced the anti−shear properties of the C/C bolt but did not vary the failure mechanism of the latter.

## 4. Conclusions

A continuous and dense SiC/Si coating was formed on C/C bolts by silicon infiltration modification. The coating was distributed continuously and densely. The SiC−Si coating was tightly bonded with bolts through the pinning effect. There was no obvious delamination, and the coating filled the pores of C/C threads.The mechanical properties of C/C bolts after silicon infiltration modification were significantly enhanced, exhibiting a tensile strength of 55.16 MPa and a double−sided shear strength of 54.73 MPa. Compared with unmodified C/C bolts, the tensile strength and double−sided shear strength of the modified bolt increased by 26.83% and 24.73%, respectively. The tensile failure of C/C−SiC bolts was caused by the breakage of studs, and the double−sided shear failure was induced by the crushing of threads and shear fracture of studs. Under tensile stress, the tensile load was transferred from threads to studs through the coating, and the failure mode changed from the pull−out of threads to the breakage of studs. This provided the fiber−toughening mechanism and increased the load−bearing capacity of the bolt.The failure mechanism of the C/C bolts did not vary after silicon infiltration modification. C/C and C/C−SiC bolt failures consisted of matrix cracking, fiber debonding, and fiber bridging. The SiC and C coating formed by silicon infiltration modification could effectively bear and transmit loads to the internal C matrix and C fiber, thus exerting the reinforcing and toughening effect on the bolts.

## Figures and Tables

**Figure 1 materials-16-01785-f001:**
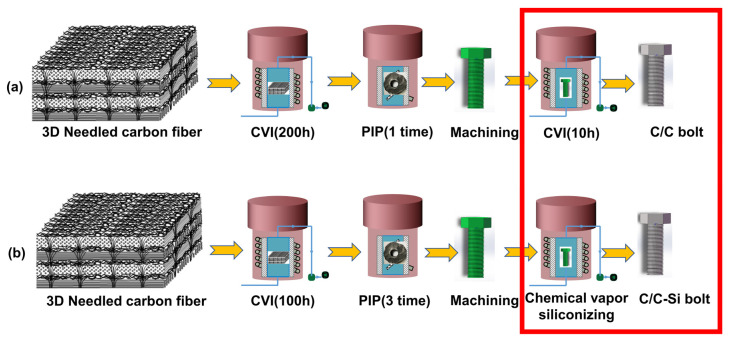
Flow chart of bolt preparation. (**a**) C/C bolts; (**b**) C/C−SiC bolts.

**Figure 2 materials-16-01785-f002:**
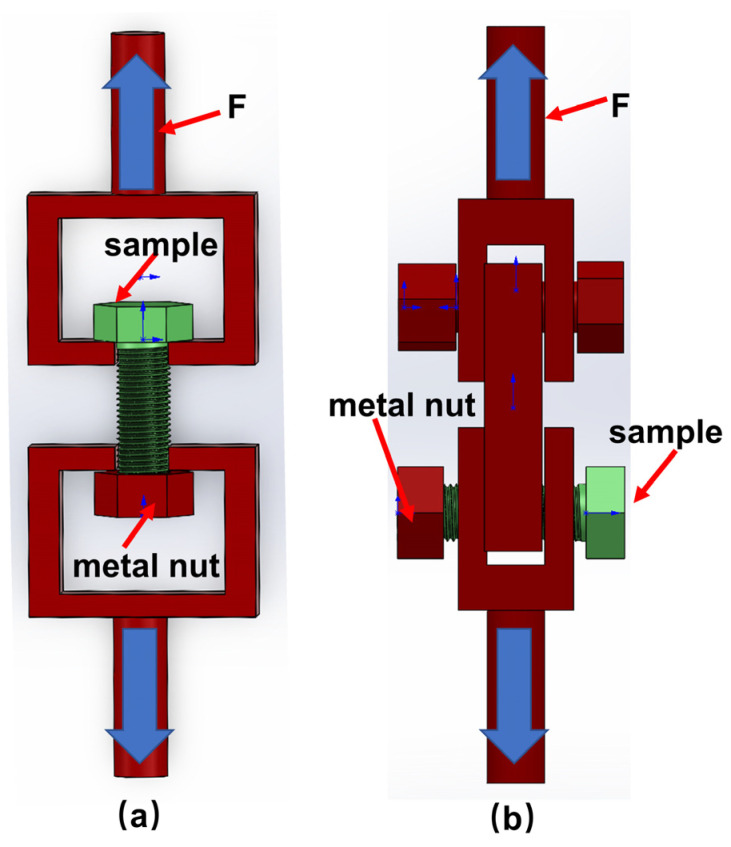
Bolt mechanical properties test chart. (**a**) tensile; (**b**) double−sided shear.

**Figure 3 materials-16-01785-f003:**
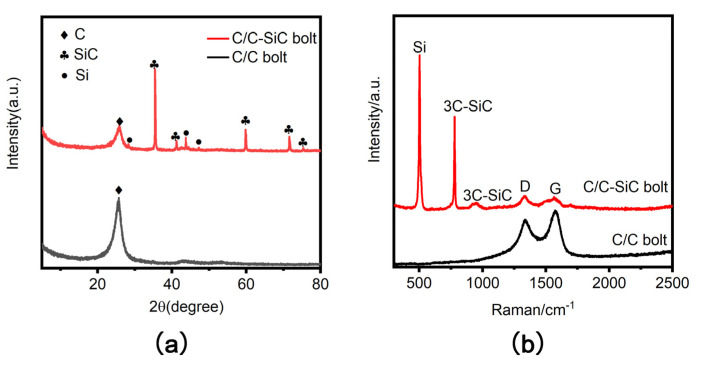
Physical phase analysis of bolts. (**a**) XRD diagram; (**b**) Raman diagram.

**Figure 4 materials-16-01785-f004:**
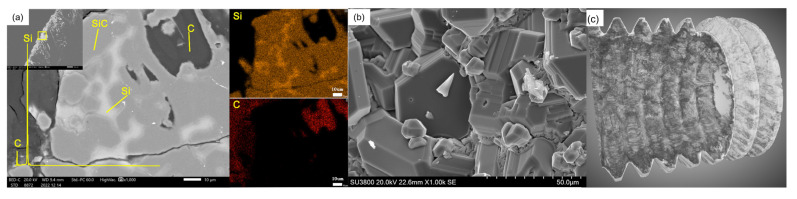
Coating morphology. (**a**) EDS image of coating; (**b**) coating morphology; (**c**) CT photograph of threaded tooth.

**Figure 5 materials-16-01785-f005:**
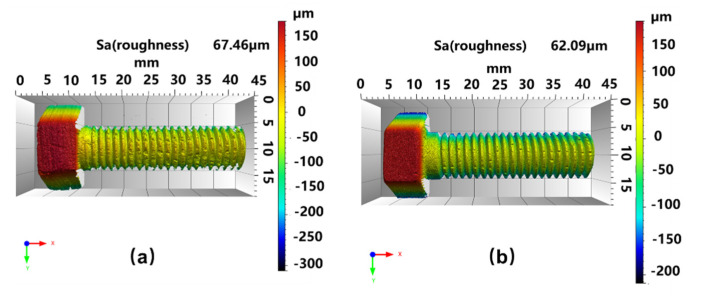
The contour diagrams of bolts. (**a**) C/C bolt; (**b**) C/C−SiC bolt.

**Figure 6 materials-16-01785-f006:**
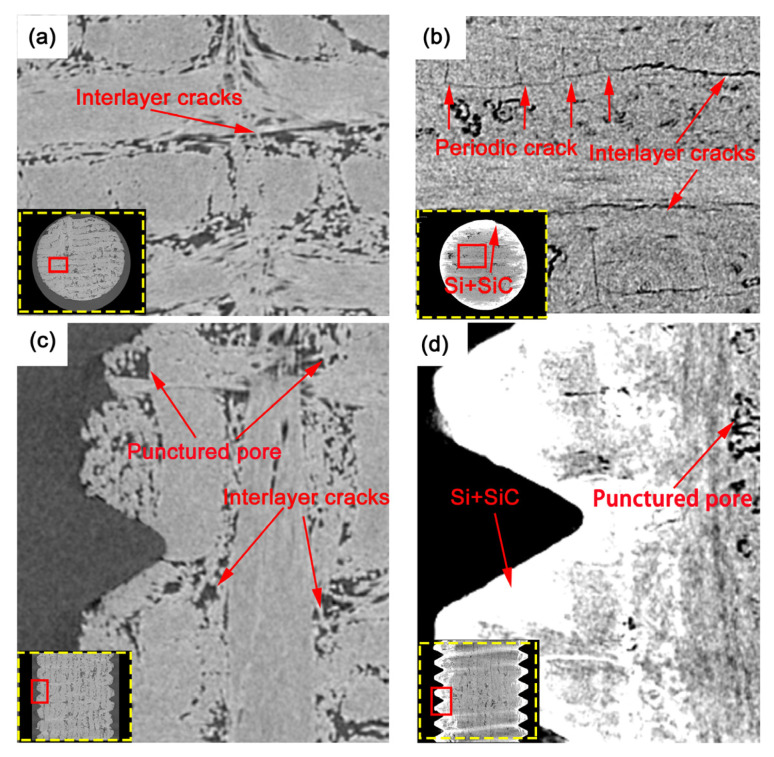
CT diagram of bolts. (**a**) ZY plane CT diagram of C/C bolt; (**b**) ZY plane CT diagram of C/C−SiC bolt; (**c**), X axial CT diagram of C/C bolt; (**d**) X axial CT diagram of C/C−SiC bolt.

**Figure 7 materials-16-01785-f007:**
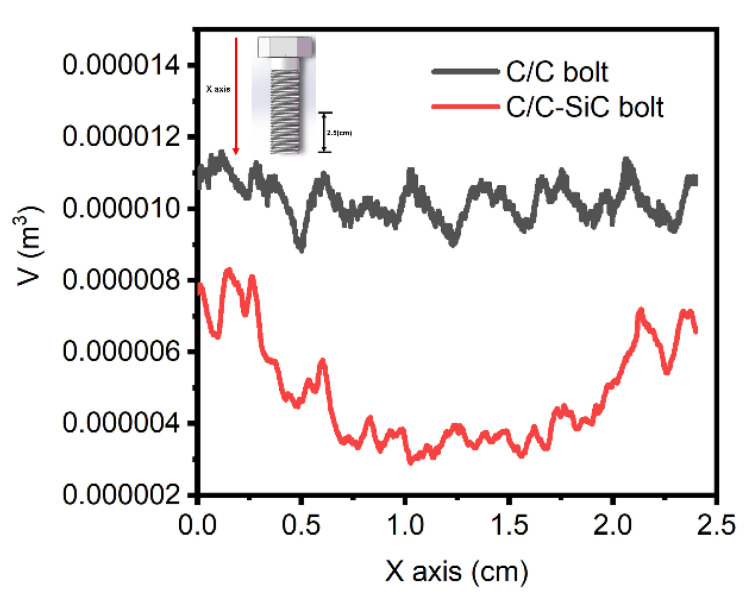
Pore volume distribution of the bolt on the *x*−axis.

**Figure 8 materials-16-01785-f008:**
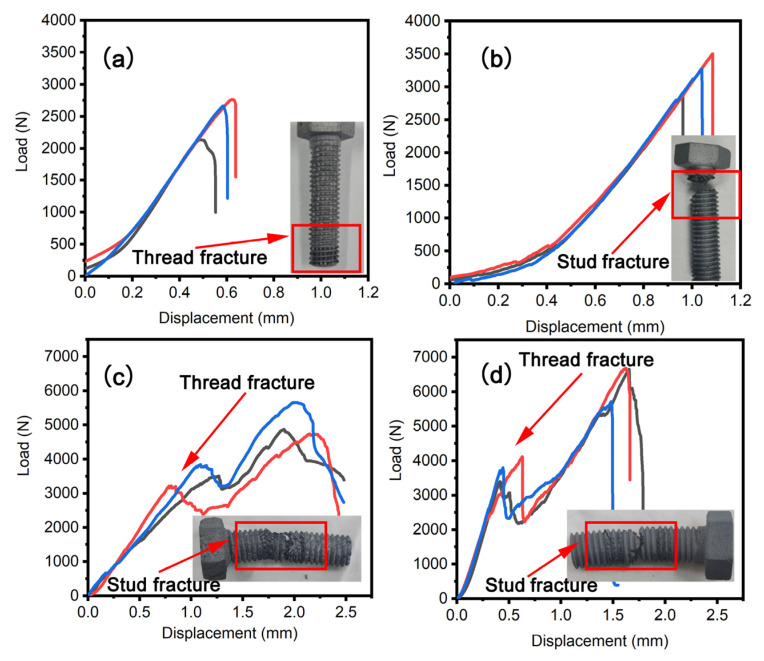
Load displacement curve of bolts. (**a**) Tensile load curve for C/C bolts; (**b**) Tensile load curve for C/C−SiC bolts; (**c**) Double−sided shear load curve for C/C bolts; (**d**) Double−sided shear load curve for C/C−SiC bolts.

**Figure 9 materials-16-01785-f009:**
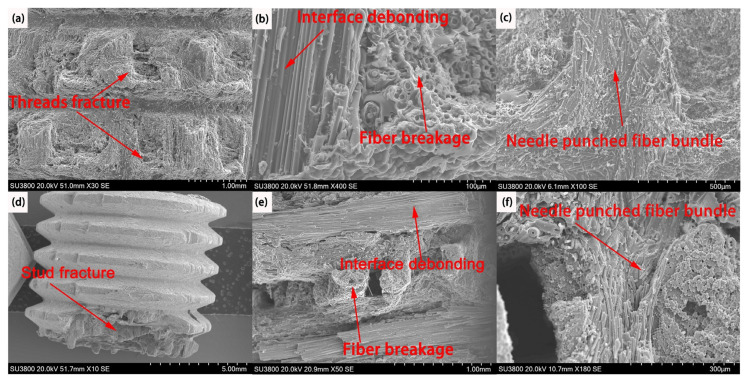
SEM image of tensile failure of bolt. (**a**) Threads fracture C/C bolts; (**b**) Interface debonding and fiber breakage of C/C bolts; (**c**) Needle−punched fiber bundle C/C bolts; (**d**) Stud fracture C/C−SiC bolts; (**e**) Interface and debonding of C/C−SiC bolts; (**f**) Needle−punched fiber bundle C/C−SiC bolts.

**Figure 10 materials-16-01785-f010:**
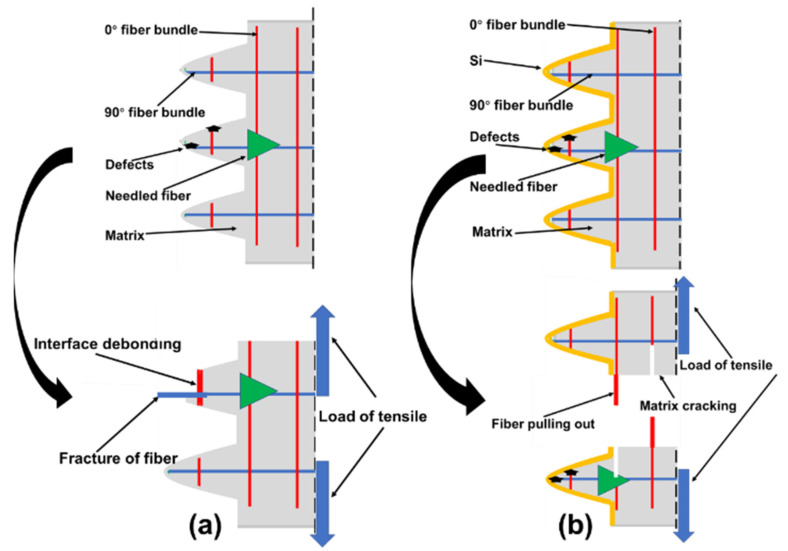
Diagram of the tensile failure mechanism of the bolt. (**a**) C/C thread failure; (**b**) C/C−SiC stud pull−out failure.

**Figure 11 materials-16-01785-f011:**
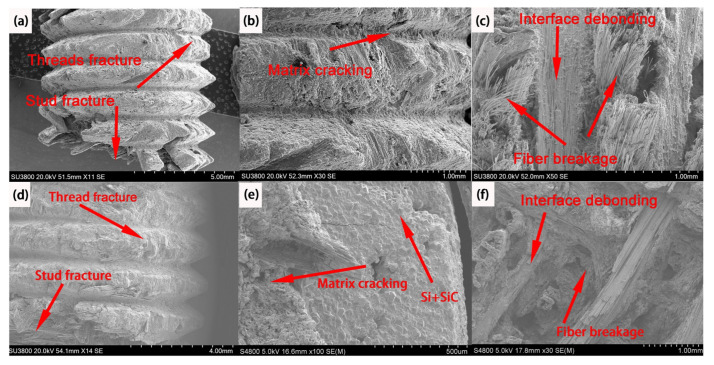
SEM image of double−sided shear failure of bolts. (**a**) Thread fracture and Stud fracture of C/C bolts; (**b**) Matrix cracking C/C bolts; (**c**) Interface debonding and fiber breakage of C/C bolts; (**d**) Thread fracture and Stud fracture of C/C−SiC bolts; (**e**) Matrix cracking C/C−SiC bolts; (**f**) Interface debonding and fiber breakage of C/C−SiC bolts.

**Figure 12 materials-16-01785-f012:**
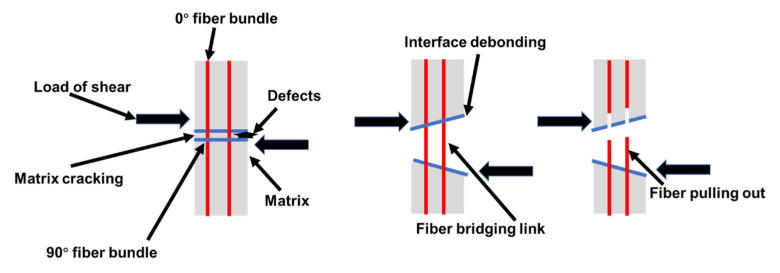
Diagram of the double−sided shear failure mechanism of the bolt.

**Table 1 materials-16-01785-t001:** Bolt mechanical properties test results.

Sample	Tensile Strength/MPa	Double−Sided Shear Strength/MPa	Compression Load/N
C/C	43.49 ± 5.80	43.88 ± 4.29	3518.67 ± 309.61
C/C−SiC	55.16 ± 5.17	54.73 ± 4.73	3770.00 ± 369.61

## Data Availability

The data presented in this study are available on request from the corresponding authors.

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
