# Peer review of "Influence of Si and SiC Coating on the Microstructures and Mechanical Properties of C/C Bolts"

_materials, 2023, doi:10.3390/ma16051785_

Round 1

Reviewer 1 Report

The manuscript “Microstructure and Mechanical Properties of Silicon Infiltration Modified C/C Bolts” describes the preparation of C/C bolts and modification with silicon infiltration (SiC). The microstructure and mechanical properties of the prepared samples have been studied. It was reported that the tensile and shear strength had been improved. However, many imprecise places in the manuscript need to be corrected, and revisions must be made before it can be considered for publication.

1-     The title should be modified.

2-     The resolution of Fig. 1 needs to be improved.

3-     In line 117, three references were mentioned for Fig. 3a and 3b. Do these figures belong to the cited references?

4-     Data and figures should be adequately explained.

5-     The caption in Fig. 7 should be corrected.

6-     Authors are advised to add labels of (a), (b) …and the related description in Figs 9, 10, 12, and 13.

7-     The manuscript has an excessive number of figures. The authors should consider the possibility of placing some of those figures as supplementary material.

8-     It would be necessary to see how the obtained results compare with those already published. 

Reviewer 2 Report

The article is written properly and investigations were done appropriately

Few concerns 

Why the bolt of this specific size is selected ? cant find any reason for this except for aerospace application, it can be of different size, please justify?

 Literature regarding how bolts fail in different modes, shear and tear and in combination should be added. Later comparison of recent work with metallic counterpart in discussion should be included

Materials and methods explained well. Have authors checked, if PIP cycles increased from 3 to 5, what will happen? How its optimization being carried out ?

bolt were 94 measured using a hydraulic servo electronic universal tester (Changchun DNS-100) at a 95 speed of 0.5 mm/min.” Why speed of 0.5mm/min, if some standard followed, please mention otherwise explain it

In figure 8, comparison should be added with conventional bolts being used in aerospace industry, so that a fair comparison can make study more a worth

Secondly, why the shear lod curve has a kink, explain that pseudo- behaviour

Fig 9, interface bonding where it is highlighted is not clear ?

Fig 10,11, 12, explained well

Conclusion part, especially point need, the research bolt comparison with its conventional bolts 

Reviewer 3 Report

In the presented work, an attempt was made to enhance the properties of carbon/carbon composite bolt by silicon infiltration and analyze the microstructure and mechanical properties of the enhanced materials. The manuscript is well organized and written. The paper is of appropriate length. The title and abstract are satisfactory. The figures and tables are appropriate and informative. I found the approach and conclusions to be robust and useful. The number of bibliographic references is sufficient. The state-of-the-art review presented in the Introduction part is comprehensive and follows a good logical structure. The testing guidelines and equipment used for carrying out the experiments are fully provided, and the obtained results are thoroughly presented and discussed accordingly. Finally, the Conclusions part does a good job in wrapping up the paper by summarizing the main findings. However, here are some comments and suggestions which can help improve the quality of the manuscript.

·       Figures need improvement. For example, in Figure 1, the image is not clear.

·       As reported by the authors, thermal stress was developed due to the high-temperature silicon infiltration. This resulted in cracks at the interlayer. Does this have any influence on the evaluated mechanical properties?

·       The tensile strength of C/C bolt increased by 26.83% after infiltration. It would be helpful to the readers if the authors can state the reason for this improvement

·       The authors should inform to the audience what is novel in the work carried out.

·       The authors are requested to do a thorough proofreading to improve the quality of the manuscript
